# Effects of sudden stratospheric warmings on the global ionospheric total electron content using a machine learning analysis

Guanyi Ma[1], Klemens Hocke[2,3]

[1]National Astronomical Observatories, Chinese Academy of Sciences, Beijing 100101, China

[2]Institute of Applied Physics, University of Bern, 3012 Bern, Switzerland

[3]Oeschger Centre for Climate Change Research, University of Bern, 3012 Bern, Switzerland

*Correspondence to*: Guanyi Ma (guanyima@nao.cas.cn), Klemens Hocke (klemens.hocke@unibe.ch)

**Abstract.** A sudden stratospheric warming (SSW) is a breakdown of winter stratospheric polar vortex. It has atmospheric effects in both the Northern and Southern hemispheres, leading to disturbances in the whole ionosphere. Previous works with

case studies have shown that SSW effects are observed mainly in low-latitude ionosphere and each SSW event may have a different effect on the ionosphere due to complex dynamics from solar/geomagnetic activities and seasonal changes. However, the SSW induced tidal variability in mid to high-latitude ionosphere is only identified for several events and its behaviour is not well understood. Here we analyze major SSWs' influences on diurnal/semidiurnal variations of global ionosphere with the global maps of total electron content (TEC) from 1998 to 2022. We use machine learning (ML) with

neural network to establish the TEC (ML-TEC) model related to the solar/geomagnetic activities and seasonal change from the long-term global TEC data. The TEC variations due to SSWs are extracted by subtracting the ML-TEC from the observed TEC. Comprehensive composite analysis of 18 major SSW events shows for the first time a globally SSW-induced enhancement in diurnal/semidiurnal TEC variations. The enhancement is the strongest at equatorial ionospheric anomaly (EIA) crests, moderate in mid-latitude and vague in high-latitude ionosphere. It also exhibits hemispheric asymmetry and

longitudinal differences. While the semidiurnal enhancement starts earlier and peaks at ~8 days after SSW onset, the diurnal one starts on the SSW onset day and peaks around 20-30 days after SSW onset. The enhancement of both semidiurnal and

diurnal TEC variations lasts for about 20~50 days after SSW onset. The SSW related E-region dynamo is likely the dominant mechanism which is not strong enough to produce discernible TEC variations in high-latitude ionosphere. ML-TEC does not contain the SSW effect and is thus a valuable reference for the ionospheric state without an SSW.

## 1 Introduction

A sudden stratospheric warming (SSW) event is associated with a breakdown and reversal of the stratospheric polar vortex of the winter hemisphere. This severe disturbance of the vortex is caused by the interaction of upward propagating planetary waves and the stratospheric zonal mean wind during the winter months. SSWs influence the atmosphere above the stratosphere by causing widespread effects on atmospheric chemistry, temperatures, winds, neutral particles, electron densities, and electric fields. They also have great atmospheric effects in the hemisphere opposite from the location of the original SSW, causing changes in the whole atmosphere and ionosphere (Pedatella et al., 2018; Baldwin et al., 2021, Goncharenko et al., 2022). Although the specific definition of SSWs has varied over years, it is now widely accepted that a major SSW event mainly occurs in the winter period of the Northern Hemisphere. It is manifested by the reversal from eastward to westward of the zonal mean wind at 10 hPa and 60 °N and the increase of the stratospheric temperature in the polar region (Goncharenka et al., 2021).

The definitive observation of the ionospheric variations due to SSWs was first reported by Goncharenko and Zhang (2008) and Chau et al. (2009). Since these studies, the effects of the SSWs on the ionosphere have been considerably conducted with both observation and simulation. The underlying mechanism can be modified E-region dynamo for ionosheric effects observed at low to mid-latitudes. It has been established through multiple simulations that an SSW induces wind and temperature changes in the middle atmosphere. The upward propagating atmospheric tides from below are often amplified

by these changes and induce stronger electric field variations in the ionospheric dynamo region during an SSW. The electric field variations at low and middle latitudes are mapped via the magnetic field lines into the low latitude F region where E x B plasma drifts lead to considerable changes of the equatorial plasma distribution during an SSW (Jin et al., 2012; Pedatella and Liu, 2013; Pedatella et al., 2014). Ionospheric variations at mid-latitude are also explained by changes in F-region thermospheric wind, combination of tidal disturbances in thermospheric wind and electric field, and upwelling in changed $O/N_2$ thermospheric composition caused by upward-propagating solar/lunar tidal amplifications due to SSW effects on the middle atmosphere (Fuller-Rowell et al.,2010; Chernigovskaya et al., 2018; Goncharenko et al., 2021).

The majority of previous works conducted case studies to analyze the impacts of the SSW on the ionosphere. There are indications that each SSW event may have a different effect on the ionosphere due to complex dynamics from solar and geomagnetic activities (Goncharenko et al., 2021). Moreover, the effect is particularly large in the low-latitude, where a strongly amplified semidiurnal pattern in the vertical ion drift, equatorial electrojet and TEC have been observed (Chau et al., 2009; Yamazaki et al., 2012; Goncharenko et al., 2021). SSW induced tidal variability in mid-latitude ionosphere are only identified for a few events although enhancement in F-region electron density, height and temperature have been observed (Xiong et al., 2013; Chen et al., 2016; Goncharenko et al., 2018; Liu et al., 2019). In high-latitude ionosphere the discerned response to SSW is confined to decrease of peak electron density and cooling/warming of ion temperature (Kurihara et al., 2010; Yasyukevich, 2018).

There has been a lack of statistical analysis on ionospheric effects related to SSWs. The average behaviour of the SSW-induced ionospheric changes is not well understood. Recently a composite analysis of 29 major SSW events was performed with the long-term series of peak electron density ($NmF_2$) over Okinawa in the northern border of the low-latitude

ionosphere. Moderate SSW influence was found in the semidiurnal amplitude averaged across 29 major SSW events compared with that in the no-SSW years (Hocke et al., 2024a). There were several other studies that investigated response to

SSW at middle to high latitudes, including for multiple events. It has been shown that enhanced semidiurnal lunitidal (M2) perturbations extended to middle latitude in the Southern hemisphere. In the American sector around -75 °E, semidiurnal tides in the mid-latitudes of the Southern hemisphere are stronger than those in the Northern (Liu et al., 2021; 2022). However, the general effects of SSW on the tidal variability in the mid to high-latitude ionosphere has never been addressed from a statistical perspective.


This paper uses the long-term time series of global TEC to derive an average tidal/semidiurnal response of the global ionosphere to major SSWs by means of a comprehensive composite analysis. The diagnosis of the SSW effect becomes relatively straightforward since the accidental ionospheric variations during SSW events can be smoothed out. On the other hand it is crucial to quantify ionospheric disturbances driven by SSWs from the atmosphere below and to distinguish those

disturbances from solar/geomagnetic forcing above. Moreover, the seasonal change should be separated from the SSW effect. We use machine learning (ML) with neural network to extract the TEC (ML-TEC) series or model related to the solar/geomagnetic activities and seasonal change from the long-term TEC data. Then the TEC variations due to SSWs and atmospheric forcing from below can be obtained by subtracting the ML-TEC from the observation. The data and methodology are described in Section 2. Presented in Section 3 are the results of data analysis. Discussion is in Section 4 and

conclusions are given in Section 5.

## 2 Data and Methodology

The ionospheric vertical TEC (just referred to as TEC in this paper) can be derived by using the dual-frequency measurements from Global Navigation Satellite System (GNSS) ground receivers due to the dispersive characteristics of the ionosphere. With ~300 GNSS stations distributed worldwidely, the International GNSS Service (IGS) has routinely provided global ionospheric maps (GIMs) of TEC (GIM-TEC) with a time resolution of 2 h and a spatial resolution of 5 ° in longitude and 2.5 ° in latitude since 1998. The map has 71x73 grid points in latitude and longitude. Details on the derivation and evaluation of the GIM-TEC were described by Hernández‐Pajares et al. (2009). Accumulated more than two solar cycles, the long term dataset of global TEC has been used for construction of ionospheric TEC model, analysis of climatological characteristics of the ionosphere and space weather. Recently IGS GIMs have been used to study lunar tides in the ionosphere (Pedatella, 2014; Hocke et al., 2024b). The GIM-TEC used in this paper is from 1998 through 2022. It should be pointed out that the GNSS stations are unevenly allocated, especially in earlier periods. Over vast oceanic regions near the equator GNSS receivers were sparsely set up on islands where adjacent receivers separated by a longitude difference up to 20 degrees. There were no receivers in the Southern hemisphere high latitudes around 120 °W over the Western Pacific Ocean and 15 °W over the Atlantic Ocean (Schaer, 1999). Additionally the inclination of GNSS satellites inherently limits the satellite visibility at high latitudes near the polar region. In areas lacking observation the TEC retrieval inevitably involves interpolation, which can affect the accuracy. Therefore, our analysis focuses on low and mid-latitudes, where GNSS data is more reliable.

There are 18 major SSW events from 1998 to 2022 and all of them happened in the Northern hemispheric winter (Hocke et al., 2024a). Table 1 presents the SSWs in their central date, which is also referred to as SSW onset hereafter. The central date of each SSW event is determined by the time when the zonal mean wind changes from eastward to westward at 10 hPa,

northward of 60 N (Palmeiro et al., 2023; Vargin et al., 2022). The events dated 20100323 and 20220322 occurred later in the season. They could be classified as Final Warmings. However, they were included in our analysis because they met the criteria for major SSWs as defined by Goncharenko et al. (2021).

Table 1. Central dates of the 18 SSW events from 1998 to 2022 in the Northern hemisphere.

| 19981215 | 19990225 | 20010211 | 20011230 | 20020217 | 20030118 | 20040105 | 20060120 | 20070224 |
| 20080222 | 20090124 | 20100209 | 20100323 | 20130106 | 20180211 | 20190101 | 20210104 | 20220322 |

The primary factor that determines the TEC is solar extreme ultraviolet radiation. The solar radio flux at 10.7 cm (F10.7) and Lyman-alpha (Lα) are generally used as proxies for the solar activity. Deviations of the ionosphere from its background can be caused by geomagnetic disturbances. Kp index is a globally averaged indicator of the worldwide level of geomagnetic

activity. Day of year informs about the seasonal change in the atmosphere. These four kinds of data are used as driven parameters to quantify the TEC variations associated with solar, magnetospheric and seasonal variations. We use machine learning with a multilayer feed-forward neural network (MFFNN) to construct the ML-TEC model from the GIM-TEC. The MFFNN consists of the input layer, two hidden layers and the output layer. A schematic diagram of data flow in the network is shown in figure 1. The input layer has 8 nodes. F10.7 and Lα for solar activity, Kp for geomagnetic activity, Kp(-3 d)

for 3-day delayed geomagnetic activity, $cos(2\pi\frac{h}{24})$ and $sin(2\pi\frac{h}{24})$ represent the diurnal variation in ionospheric TEC due to the earth rotation, $cos(2\pi\frac{DOY}{365})$ and $sin(2\pi\frac{DOY}{365})$ for the earth revolution, $cos(2\pi\frac{DOY}{180})$ and $sin(2\pi\frac{DOY}{180})$ are considered for the seasonal variation in the ionosphere since the central dates of the SSW events are in the northern hemispheric winter. The number of nodes in each hidden layers is 30. The output layer is the modeled TEC ($TEC_m$) from the neural network. The network is trained by backpropagation by using an approximate steepest decent rule to

minimize the squared residual error of the $TEC_m$ and fine-tune the weights (Hagan and Menhaj, 1994). We consequently

obtain the ML-TEC model determined by the solar/geomagnetic activities and seasonal change.

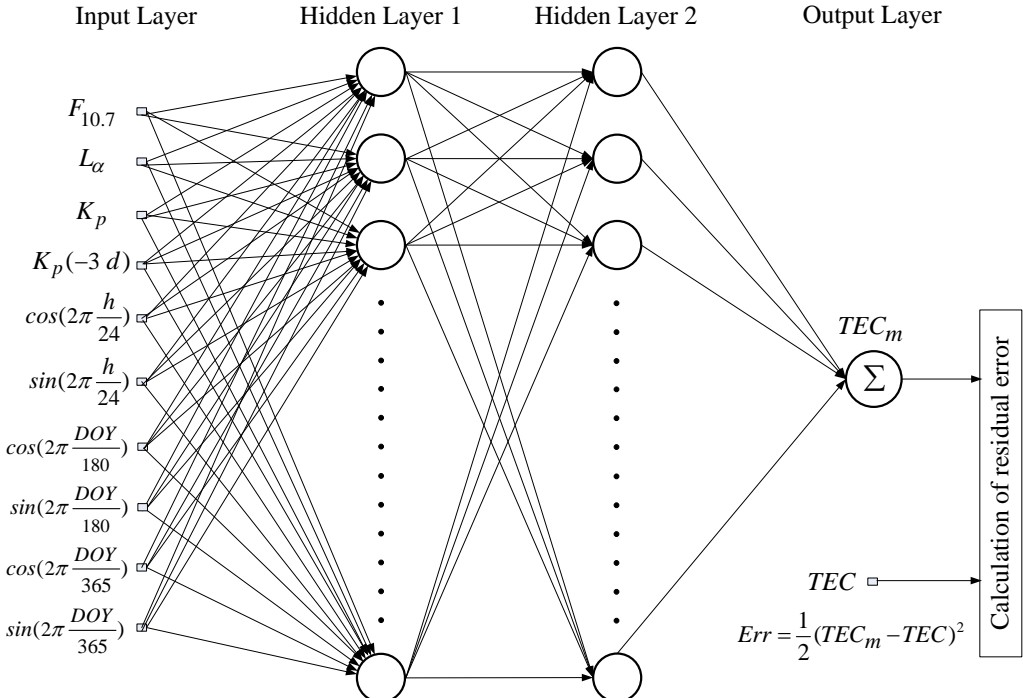

Figure 1. Schematic diagram of the MFFNN for modeling ionospheric TEC in association with solar/geomagnetic activities

and seasonal change.


The ML-TEC model fits to the global TEC observation with a zero systematic error and a root mean squares error (RMSE)

of 2.8 TECU. This is comparable to the zero systematic error and the RMSE of 3.4 TECU for the empirical function

modeling with the global TEC from 1999 to 2011 in Mukhtarov et al. (2013), and the RMSE of 3.5 TECU for a statistical

model established by Lean et al. (2016) with the global TEC from 1998 to 2015. Figure 2 presents the global maps of the

modeled and observed TEC in geographical coordinate. The equatorial ionospheric anomaly (EIA) locates between 22.5 °S

and 25 °N around 105 °E, with the summer crest being stronger than the winter one. The Weddell Sea Anomaly is apparent

with the stripe amplification between 80 ˚S to 50 °S and -120 ˚E to 0 °E (Mukhtarov et al., 2013). The coincidence of these

anomalies indicates the ML-TEC model is also able to reproduce the spatial structure of the ionosphere.

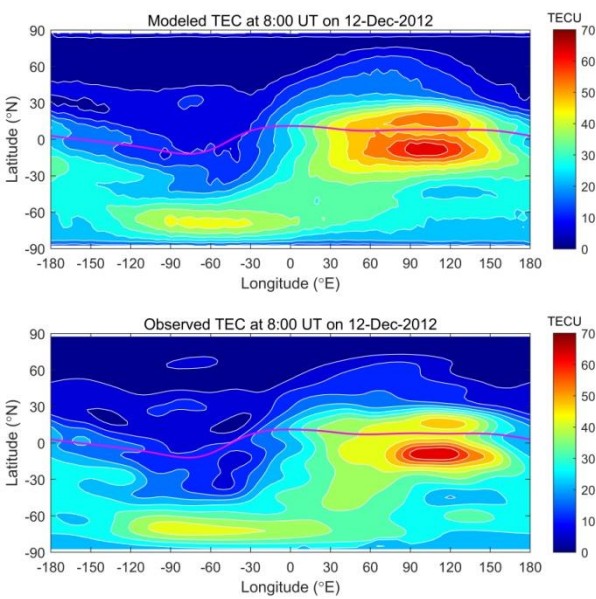

Figure 2. Global maps of the modeled and observed TEC at 0800 UT on 12 December 2012. The lines in magenta represents

the magnetic equator.

The diurnal ( $s_1$ ) and semidiurnal ( $s_2$ ) components in TEC time series are obtained with a digital non-recursive, finite-

impulse-response (FIR) filter. It performs zero-phase filtering by processing the time series in forward and reverse directions

which helps preserve features in a filtered time waveform exactly where they occur in the unfiltered signal. For band-pass

filtering, the cutoff frequencies are at $f_c = f_p \pm 10\% f_p$, where $f_c$ is the cutoff frequency and $f_p$ fp is the central

frequency. For the diurnal and semidiurnal variations, the cutoff frequencies are 0.9/1.1 and 1.8/2.2 cycles per day (cpd),

respectively (Hocke et al., 2024a; Studer et al., 2012).

For the 18 SSW events listed in Table 1, by using the time series of TEC with 2 h resolution, 18 subsets of the observed and

modeled global TECs are created that started 200 days before the central date of SSW and ended 200 days after. The

flowchart of the further data processing is shown in figure 3. For each SSW event, $s_1$ and $s_2$ of both the observed and

modeled TECs are extracted by applying the FIR filter to the corresponding dataset. They are referred to as $s_{1o}$, $s_{2o}$, $s_{1m}$

and $s_{2m}$, respectively. With diurnal and semidiurnal components of 18 SSW events, the composite analysis calculates the

mean of $S_1$ and the mean of $S_2$ for the observed and modeled TECs, represented as $S_{1o}$, $S_{2o}$, $S_{1m}$ and $S_{2m}$, respectively.

It can be expected that an inherent effect of SSW can be seen well in the mean values while accidental variations contributed

to $S_1$ and $S_2$ compensate one another. Then the difference of the composites between observation and model is taken,

expressed as $\Delta S_1$ and $\Delta S_2$. This operation removed the solar/geomagnetic and seasonal effects in the diurnal and

semidiurnal components and only those driven by the atmosphere below are retained. As shown by $rS_1$ and $rS_2$, the ratios

of those observed to the modeled ones are also calculated to show the relative strength of SSW-related disturbances.

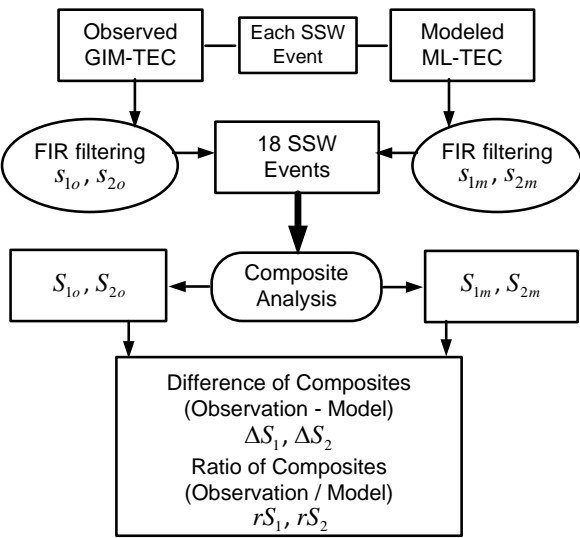

Figure 3. Flowchart of FIR filtering of GIM TEC and ML-TEC, and composite analysis of tidal TEC variations for the 18

SSWs.

## 3 Results

We begin with a case study for the major SSW event of 25 February 1999 as shown in figure 4. The top panel presents the

diurnal components at grid point (30 °N, 105 °E) for both the observed GIM-TEC (black line) and modelled ML-TEC (red

line). The middle panel is for the semidiurnal components at grid point (30 °N, 105 °E). The bottom panel gives F10.7 and Kp

indices to show the solar and geomagnetic conditions during the event. The epoch time is from 50 days before the onset of

SSW and 100 days after the onset of SSW. The two $s_1$ time series start to increase around SSW onset although they are

close to each other and oscillate together in the preceding time. The $s_1$ of the observed TEC is smaller than the model one

before SSW onset. However, the $s_1$ of the observed TEC becomes lager since ~2 days and shows a maximum at an epoch

time of 20 days, which is ~2.5 TECU larger than the modeled one. The $s_1$ of the observed TEC keeps larger than that of the

ML-TEC for about ~30 days. The $s_2$ of the observed TEC varies in anti-phase with that of the modelled TEC before SSW

onset, It starts to be larger than that of the modeled TEC at ~10 days, and reaches a maximum at 20 days. The largest

difference is ~1.6 TECU between the observed and modeled ones. The $s_2$ of the observed TEC keeps larger than that of the

modelled TEC for ~80 days. Note that $s_1$ from the modelled TEC correlate more with F10.7 variation while there is no

obvious variation for both $s_1$ and $s_2$ corresponding to geomagnetic activities.

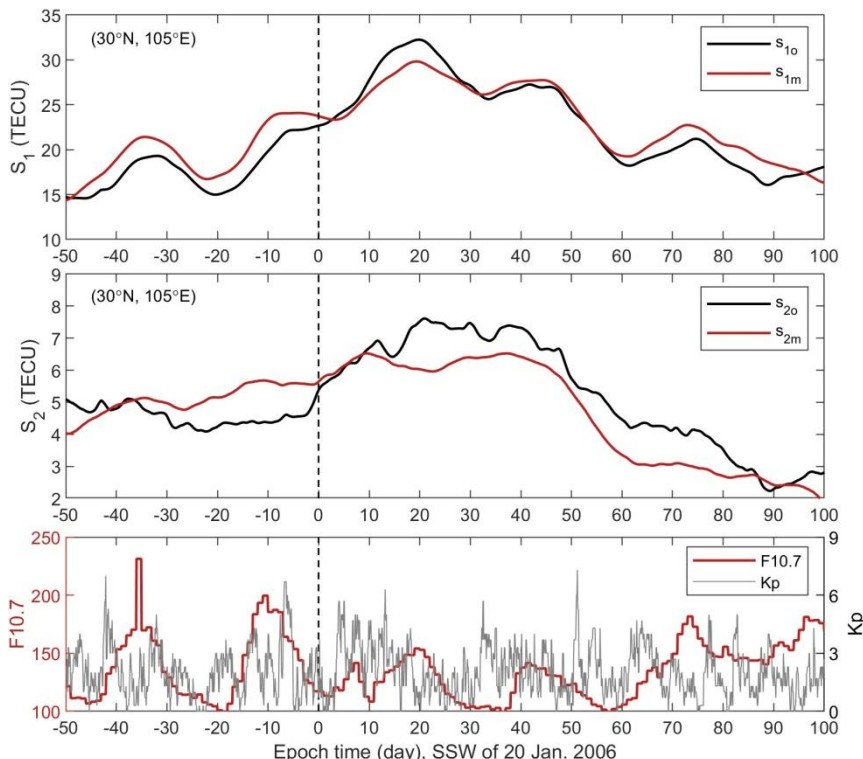

Figure 4. Diurnal (top panel) variation and semidiurnal variation (middle panel) of TEC at (30 °N, 105 °E) (top panel) from observed TEC (black lines) during a major SSW event centered on 25 Feb. 1999 (marked as the vertical grey lines). Together shown are the diurnal and semidiurnal components from modeled TEC (red lines) for the event. The solar and geomagnetic conditions during the event are displayed by F10.7 (red line) and Kp (grey line) (bottom panel).

Results of composite analysis are shown by world maps, latitude-time and longitude-time plots. Since both the diurnal and semidiurnal components vary with latitude, longitude and time, we select world maps of those with overall smallest values before SSW onset and largest values after SSW onset. For clarity in our description and discussion, we use hereafter the subscript "b" to denote the period before the SSW onset, and the subscript "a" to denote the period after the SSW onset. Figure 5 exhibits world maps of diurnal variation from composite analysis of the 18 SSW events at 13 days before SSW onset and 25 days after SSW onset. The thick black line in each map depicts the magnetic equator. In the left panel $\Delta S_{1b}$

of (a) is the global $\Delta S_1$ at 13 days before SSW onset. Conspicuous negative $\Delta S_{1b}$ occupies most of the land areas. Positive $\Delta S_{1b}$ locates mainly over oceans and high latitudes to polar regions. However, it can be spotted over lands in the mid-latitudes of the Northern hemisphere and the low latitudes in the Southern hemisphere. Shown in Figure 5b, at 25 days after SSW onset, conspicuous positive $\Delta S_{1a}$ prevails almost globally. The largest values locate around the EIA in the Northern hemisphere. It appears with moderate strength around 90°E and 75°W in mid to high latitudes in the Northern hemisphere, and in mid to high latitude for all longitudes in the Southern hemisphere. There are also positive patches in low latitudes in south America and its west. Contrasting $\Delta S_{1a}$ and $\Delta S_{1b}$, the amount of change, referred to as, $\Delta S_{1E}$ can be obtained by taking the difference, $\Delta S_{1a} - \Delta S_{1b}$. An enhancement (positive $\Delta S_{1E}$) can be discerned globally. The enhancement larger than 1.0 TECU mainly distributes in low northern latitudes in a longitude range of [135°W, 100°E]. The largest $\Delta S_{1E}$ is 1.5 TECU and locates at (5°N, 85°W). In the Southern hemisphere, $\Delta S_{1E}$ can be larger than 1.0 TECU from low to high latitudes around 75°W in American sector. Therefore the enhancement is generally stronger in the Northern hemisphere than the Southern one. The right panel of Figure 5 is for $rS_1$ with (c) at 13 days before SSW onset and (d) at 25 days after SSW onset. The $rS_1$ larger than 1 matches positive $\Delta S_1$, and the $rS_1$ smaller than 1 to negative $\Delta S_1$. Similar spatial distributions can be noticed to those of $\Delta S_1$ by comparing the corresponding maps in the left panel. At 25 days after SSW onset $rS_{1a}$ is also stronger in the Northern hemisphere than the Southern hemisphere. However, it has a similar level at the Northern low and mid-latitudes. Note that largest $rS_1$ locates at high latitudes near polar regions, which is different from that of $\Delta S_1$. This can be attributed to the small values of diurnal variation due to the smaller TEC there than low to mid-latitudes.

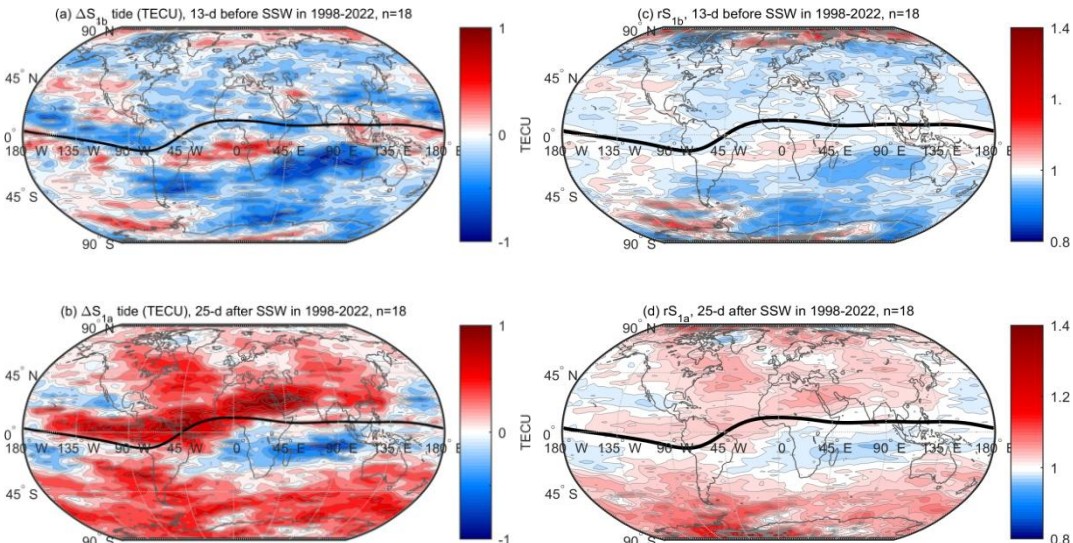

Figure 5. Distributions of $\Delta S_1$ and $rS_1$ from composite analysis of the 18 SSW events at 13 days before SSW onset and 25 days after SSW onset with (a) for $\Delta S_{1b}$, (b) for $\Delta S_{1a}$, (c) for $rS_{1b}$ and (d) for $rS_{1a}$. The thick black line in each map depicts the magnetic equator.

The global distributions of semidiurnal TEC variation are shown in figure 6 with (a) the global $\Delta S_{2b}$ at 12 days before SSW onset and (b) $\Delta S_{2a}$ at 8 days after SSW onset in the left panel, and those of $rS_2$ in the right panel. At 12 days before SSW onset $\Delta S_{2b}$ is generally between -0.5 and 0.8 TECU in magnitude. In the Northern hemisphere, positive $\Delta S_{2b}$ are manifested as patches at low latitudes along the magnetic equator, although a few larger patches can be seen at mid and high latitudes. In the Southern hemisphere $\Delta S_{2b}$ is more active with red patches or belts fill at low to mid-latitudes and high latitude in the American, Atlantic and Asian sectors. At 8 days after SSW onset $\Delta S_{2a}$ peaks along the magnetic equator at EIA in both hemispheres with the largest value of 2.0 TECU around (30 S, 85 W). The red belt is generally wider in the Southern hemisphere than the Northern one. By contrasting $\Delta S_{2a}$ and $\Delta S_{2b}$, $\Delta S_{2E} = \Delta S_{2a} - \Delta S_{2b}$, we can perceive an enhancement in the Northern hemisphere in most areas except Russia, east Europe, central of North America

and Pacific Ocean at ~45 °N. In the Southern hemisphere strong enhancement is more widespread and only several small white patches can be seen with much smaller areas. In land areas the largest $\Delta S_{2E}$ is 1.5 TECU at (32.5 °S, 80 °W) though it can reach 1.8 TECU around ±~20 °N over Pacific Ocean. The right panel of Figure 6 shows $rS_2$ with (c) at 12 days before SSW onset and (d) at 8 days after SSW onset. There are also similar relationships between $rS_2$ and $\Delta S_2$ by comparing the corresponding maps in two panel. At 12 days after SSW onset, $rS_{2a}$ is obviously stronger in the Southern hemisphere than the Northern hemisphere.

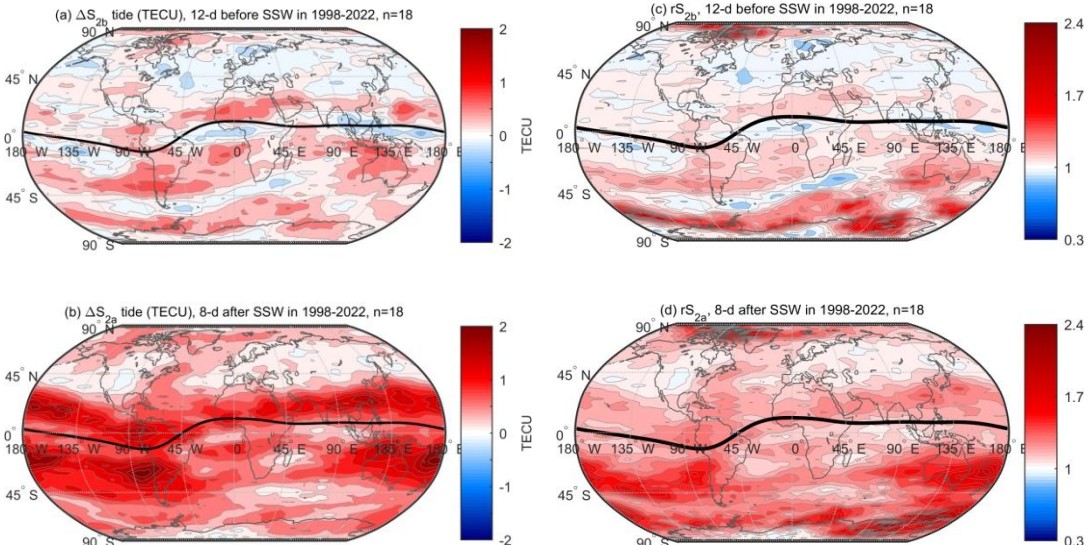

Figure 6. Distributions of $\Delta S_2$ and $rS_2$ from composite analysis of the 18 SSW events at 12 days before SSW onset and 8 days after SSW onset (b) with (a) for $\Delta S_{2b}$, (b) for $\Delta S_{2a}$, (c) for $rS_{2b}$ and (d) for $rS_{2a}$. The thick black line in each map depicts the magnetic equator.

It is important to examine the ionospheric tidal variabilities at different longitudes over time. We select two longitudes of 90 °E and 75 °W to examine the temporal variations of $\Delta S_1$ and $\Delta S_2$. As shown above by Figures 5 and 6, at ~90 °E there is

obvious enhancement of diurnal component in northern mid-latitudes; at 75 °W prominent enhancement of both diurnal and semidiurnal can be seen in both hemispheres. Figure 7 shows the time variation of $\Delta S_1$ and $\Delta S_2$ at 90 °E, which is smoothed as a 14 d average. An enhancement of $\Delta S_1$ can be seen from the SSW onset to ~35 days after the SSW onset in the whole northern latitudes. At ~25 °N the prominent positive $\Delta S_1$ starts to appear simultaneously at the onset day of SSW

235 and ends at ~40 days. The strongest enhancement happens from ~15 to ~35 °N. At ~25 °N, $\Delta S_1$ shows a peak level of ~0.6 TECU around 10 days, and maintains the highest level from 0 to ~35 days after the SSW onset. Concerning the semidiurnal component, clear enhancement of $\Delta S_2$ starts just from the SSW onset in the Northern hemisphere. The enhancement ends at ~35 days in mid-latitudes and ~50 days at low to mid-latitudes. The largest $\Delta S_2$ is 1.3 TECU centering ~20 °N and ~8 days after SSW onset. Note that there is no systematic enhancement during the entire SSW at 90 °E in Southern hemisphere with

240 few GNSS receivers due to ocean.

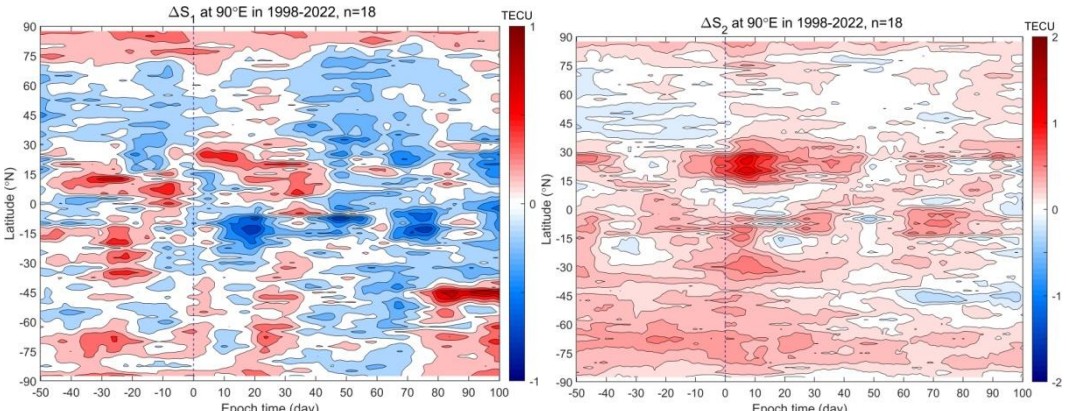

Figure 7. Time variation of meridian $\Delta S_1$ (left panel) and $\Delta S_2$ (right panel) at 90 °E, which is smoothed as a 14 d moving average. The vertical dot lines mark the SSW onset day.

245 Figure 8 is the meridian plot of smoothed $\Delta S_1$ and $\Delta S_2$ at -75 °E. It is obvious that from ~0 °N to ~70 °N an enhancement of $\Delta S_1$ starts from the SSW onset, it ends at ~50 days in low latitudes and at~30 days in mid to high latitudes.

At ~2.5 °N $\Delta S_1$ increases very fast and maintains a high level from ~8 to 22 days with a largest value of 0.8 TECU. In the southern hemisphere the enhancement of $\Delta S_1$ delays with the latitude to ~30 °S. For $\Delta S_2$ an enhancement takes place in the whole meridian from the SSW onset to ~55 days after the SSW onset except latitudes larger than 50 °S in the southern

250 hemisphere. The prominent enhancement occurs in the range from -50 °S to 40 °N, respectively. The striking positive $\Delta S_2$ starts to appear at about -10 days before the SSW onset, reaches maximum at ~8 days and ends at ~50 days after the SSW onset at both EIA regions. Note that $\Delta S_2$ has another peak at ~25 days at the northern EIA. Between ~30 °S and ~2.5 °N, $\Delta S_2$ starts to increase at ~10 days before SSW onset, reaches to its first peak of ~1.5 TECU in the period of 3 to 9 days and the second peak at ~25 days after SSW onset.

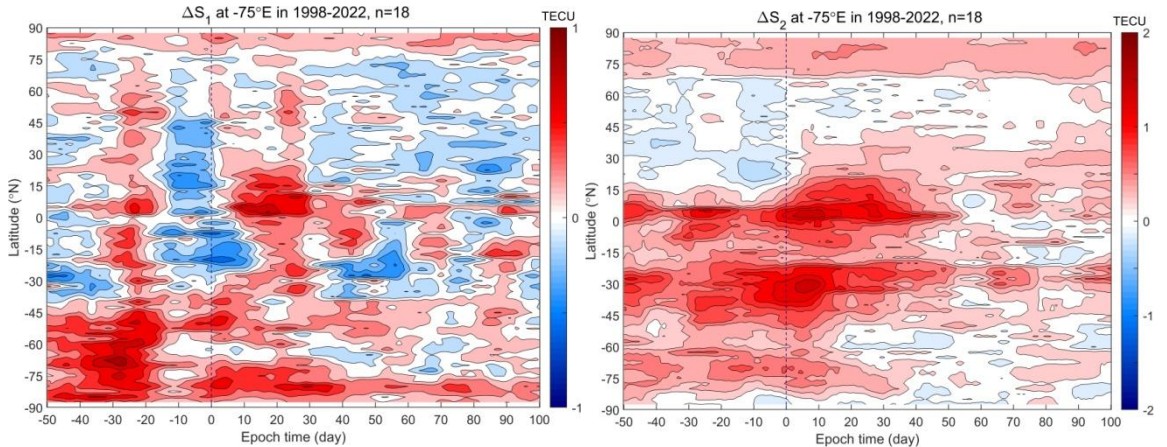

Figure 8. Time variation of meridian $\Delta S_1$ (left panel) and $\Delta S_2$ (right panel) at -75 °E, which is smoothed as a

255

14 d moving average. The vertical dot lines mark the SSW onset day.

It is also worthwhile to study the global temporal variation at specific latitude. Figure 9 plots the temporal variation of $\Delta S_1$

and $\Delta S_2$ at the latitude of 22.5 °N, which is smoothed as a 14 d moving average. The $\Delta S_1$ generally starts to increase at the SSW onset and keeps positive for ~30 days. It shows a maximum of 1.0 TECU around 20-26 days after the SSW onset. The zonal $\Delta S_2$ at 22.5 °N basically shows temporal variation synchronously although the values of $\Delta S_2$ are different at different

longitudes. It reaches a maximum of ~1.4 TECU at ~120 °E and ~8 days after the SSW onset. $\Delta S_2$ between 60-135 °E returns to the SSW onset level at ~45 days. $\Delta S_2$ between -120-60 °E decreases to the SSW onset level at ~30 days. Over the Western Pacific Ocean it returns to the onset level at ~20 days. We notice that positive $\Delta S_1$ and $\Delta S_2$ also occur and last for days before SSW onset. However, they are generally smaller, shorter-lived and slower-varying than those after SSW onset.

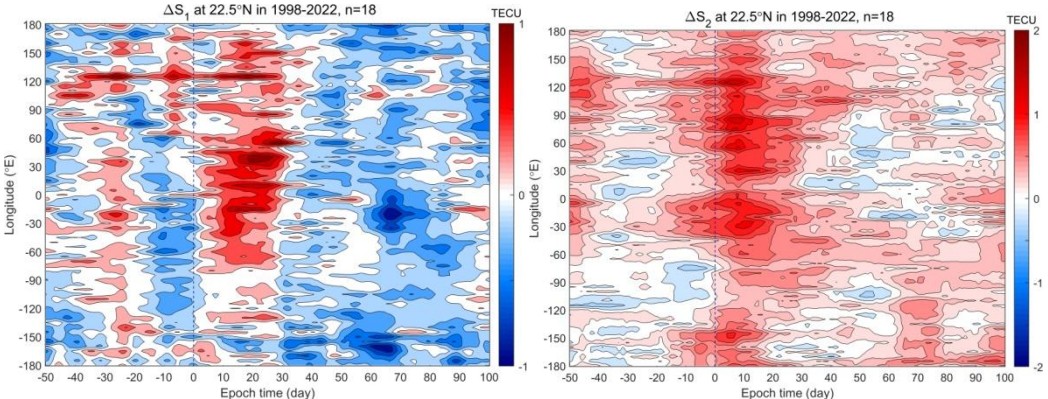

Figure 9. Time variation of zonal $\Delta S_1$ (left panel) and $\Delta S_2$ (right panel) at 22.5 °N, which is smoothed as a 14 d moving average. The vertical dot lines mark the SSW onset day.

## 4 Discussion

The driving factors of the ionosphere consist of solar and magnetospheric energies from above and the atmospheric force from below. For the attribution of ionospheric response to SSW, it is crucial to separate the atmospheric waves from effects due to solar/magnetospheric variability and seasonal variation. The case study of the SSW on 25 February 1999 in figure 4 shows intensified diurnal/semidiurnal variations of the observed TEC at 30 °N after the SSW onset. The diurnal/semidiurnal components from modelled TEC manifest contribution from solar/magnetospheric energies and seasonal change. The

comparison between the observation and model suggests a clear SSW effect on low latitude ionosphere which is in agreement with previous studies (Chau et al., 2012; Liu et al., 2019).

Since the ionosphere has local characteristics and each SSW event may have a different effect due to complicated solar-terrestrial condition, it is justifiable to perform composite analysis described in the above analysis method. The 18 SSW events happened from 1998 to 2022, which cover two solar activity cycles. The composite analysis with the solar and magnetospheric effects removed would provide unambiguous evaluation of the SSW effects on the global ionosphere.

The world maps and temporal variations over latitude/longitude of diurnal and semidiurnal components from our composite analysis in Figures 5-9 reveal that the SSW effects are indeed global as depicted by Pedatella et al., 2018. SSW-induced amplifications of diurnal/semidiurnal tides can be identified from low to high latitudes with the strongest at EIA crests along the magnetic equator. Amplifications in semidiurnal tides during SSW have been revealed in low-latitude ionosphere from both case and statistical studies (Chau et al., 2012; Goncharenko et al., 2021; Hocke et al., 2024a). Semidiurnal disturbances

in mid-latitude ionosphere have been only observed at Asian and America Sectors in Northern hemisphere (Xiong et al., 2013; Chen et al., 2016; Goncharenko et al., 2013; Liu et al., 2019). Our observation not only confirms the previous results but also displays that the semidiurnal pattern in mid-latitude ionosphere during SSWs is a global phenomenon. Interestingly, the semidiurnal enhancement is stronger in the Southern Hemisphere mid-latitude Southern hemisphere than in the Northern hemisphere. Several SSW event studies have highlighted that semidiurnal tides in the Southern hemisphere mid-latitudes,

particularly around -75°E in the American sector, are stronger than those in the Northern hemisphere. This hemispheric asymmetry may arise from the amplification of lunar semidiurnal (M2) tides during SSWs, which is the most pronounced in the American sector (Goncharenko et al., 2021; Liu et al., 2021; 2022). Additionally, the inclination angle of Earth's

magnetic field lines in the Southern hemisphere mid-latitudes is smaller than in the Northern hemisphere, leading to more ionospheric TEC variations in the F-region due to electric field effects (Goncharenko et al., 2022).


Concerning the diurnal variability, enhancement was observed at low-latitides and a mid-latitude cite of Mohe (53.5 °N, 122.3 °E) for the SSW event in 2018 (Liu et al., 2019). Our study reveals, for the first time, that diurnal TEC variations exhibit a global enhancement pattern, with stronger effects in the Northern hemisphere than the Southern hemisphere. This contrasts with the semidiurnal enhancement, which is stronger in the Southern hemisphere. Longitudinal differences are also

evident, with weaker amplification in the Atlantic, African, and Indian sectors, particularly for the diurnal tide in the Southern hemisphere. Recently, Harvey et al. (2022) emphasized the influence of the mesospheric polar vortex on atmospheric tides, which helps explain this hemispheric asymmetry. Since major SSWs occur predominantly in the Northern Hemisphere (NH) during winter, the mesospheric polar vortex in the NH significantly modulates the upward propagation of atmospheric tides to the ionosphere. This process enhances the diurnal variation of TEC, making it more pronounced in the

Northern Hemisphere.

Temporally the global diurnal component ($\Delta S_1$) at 22.5 °N starts to increase simultaneously on the day of SSW onset, peaking around 20–26 days later and lasting for approximately 30 days (Figures 9). In contrast, the semidiurnal ($\Delta S_2$) at 75 °W starts to increase simultaneously at ~10 days before SSW onset. It peaks at ~8 days and persists to ~50 days after the

SSW onset (Figure 8). At other longitudes shown in Figure 9, the semidiurnal component starts to enhance at ~20 days before SSW onset, peaks at ~8 days after SSW onset. Note that the enhancement lasts generally ~50 days during SSW with the prominent effect happens between -45 and 120 °E. These findings align with the review by Goncharenko et al. (2021), which has summarized that the main SSW effect is a distinct semidiurnal variation in thermospheric and ionospheric

parameters that lasts for days up to 30–40 days. The results of our comprehensive composite analysis for 18 SSW events

demonstrate that the enhancement of the diurnal and semidiurnal components last for ~30 and ~50 days, respectively. While

the semidiurnal enhancement starts earlier and peaks at ~8 days after SSW onset, the diurnal one starts on the SSW onset day

and peaks around 20-26 days later.

The SSW effects on the tidal ionospheric TEC variations are a global phenomenon. The complicated patterns of the SSW-

induced tidal ionospheric TEC variations indicate multiple dynamical processes might be involved during SSWs. We

speculate that the SSW related E-region dynamo is the main mechanism which is generally larger in the low-latitude

ionosphere to produce more significant vertical plasma drifts than mid-latitudes.

**5 Conclusions**

We present the comprehensive composite analysis of the ionospheric tidal variability in association with 18 sudden

stratospheric warming events by using the global total electron content data from 1998 to 2022. To extract TEC variations

from effects of SSWs and atmospheric forcing below the ionosphere, we first model the TEC climatology due to solar

activity, magnetospheric energy and seasonal change by neural network training for the observed time series of global TEC,

then we remove the modelled TEC from the observed TEC. Our analysis reveals for the first time a globally SSW-induced

enhancement in both semidiurnal and diurnal TEC variations. Key findings include:

1. Semidiurnal TEC Variations: The strongest enhancements occur at the EIA crests along the magnetic equator, consistent

with previous studies. At mid-latitudes, the semidiurnal enhancement is stronger in the Southern hemisphere than in the

Northern hemisphere, likely due to the amplification of lunar semidiurnal (M2) tides and differences in geomagnetic field

geometry.

2. Diurnal TEC Variations: Diurnal enhancements exhibit a global pattern, with stronger effects in the Northern hemisphere

mid-latitudes compared to the Southern hemisphere, contrasting with the semidiurnal enhancement.

3. Temporal Evolution: The semidiurnal enhancement starts ~10 days before the SSW onset, peaks at ~8 days after the onset,

and lasts for ~50 days. In contrast, the diurnal enhancement begins on the SSW onset day, peaks around 20–26 days later,

and persists for ~30 days.

4. Longitudinal Dependence: Both diurnal and semidiurnal enhancements show longitudinal variability, with weaker

amplification in the Atlantic, African, and Indian sectors, particularly for the diurnal tide in the Southern hemisphere.

Our analysis indicates that multiple dynamical processes might be involved during SSWs from the hemispheric

asymmetry and longitudinal differences in diurnal/semidiurnal variations of TEC. It is likely that the SSW related E-region

dynamo is the main mechanism which is generally strong enough to produce discernible TEC variations in low to mid-

latitudes ionosphere. The ML-TEC model can separate the SSW effects on the ionosphere from dependences on

solar/geomagnetic activities and season. This is a new analysis method which is important for SSW analysis since the tidal

amplitudes in the upper atmosphere have a strong seasonal dependence. The regular, seasonal enhancement of tidal

amplitudes in northern hemispheric winter can be wrongly attributed to SSWs since SSWs mainly happen in northern

hemispheric winter. Our ML-TEC model avoids such a false attribution.

**Code availability.** MATLAB codes can be provided upon request.

**Data availability.** The TEC data are a product of IGS, which are freely available at Crustal Dynamics Data Information

System, NASA's archive of space geodesy data https://cddis.nasa.gov/. The Data about solar and geomagnetic activity was

obtained from the GSFC/SPDF OMNIWeb interface at https://omniweb.gsfc.nasa.gov (accessed on 2023). The modelled

TEC data can be provided upon request.

**Author contributions.** The Concept of the study: KH and GM. Data analysis: GM and KH. Writing: GM. Corrections and discussion of the paper: KH and GM.

**Competing interests.** The contact author has declared that none of the authors has any competing interests.

**Acknowledgements.** The reviewers are thanked for their valuable comments and improvements.

**Financial support.** This research was funded by the Swiss National Science Foundation, grant number IZSEZ0-224443; the National Natural Science Foundation of China (Nos. 12073049 and 12273062), and the CAS-JSPS Joint Research Project (178GJHZ2023180MI).

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
