# Peer review of "Effects of sudden stratospheric warmings on the global ionospheric total electron content using a machine learning analysis"

_EGUsphere, 2024_

## Author Response (AR1)

To referee1

We sincerely thank the referee for their insightful and constructive comments, which have been invaluable in enhancing the quality and clarity of our manuscript. We deeply appreciate the time and effort they have dedicated to reviewing our work and providing thoughtful suggestions.

Below, we provide detailed responses (in italic font style) to each of the referee's comments (in bold face). The changed text is in a normal font style.

**Tidal changes in the ionosphere during SSW has been focused for a decade of years. Although it is believed that the changes are global, most of the studies were concerned mainly in the ionospheric variation the low latitudes. Based on the global TEC map data with resolution of 2 hour, and 2.5° latitude \*5° longitude, the ionospheric background morphology has been obtained using a neural network algorithm. Further the global distribution of the diurnal and semidiurnal tide components was analyzed using the residual of the TEC data focused on 18 major SSW events in northern winter hemisphere. This study can provide a comprehensive understanding of the global effects on the ionosphere due to SSW. On the whole, the obtained results are clear. But there are still some unclear situations and some discussions about the results are needed further.**

**Comments:**

**1, The TEC map data used here are obtained with interpolation algorithm based on about 300 GNSS stations that are very unevenly distributed. Very limited GNSS data can used in the ocean region especially in the southern hemisphere. Usually, the TEC map with such low spatial and temporal resolution are used for revealing the ionospheric background morphology and large ionosphere disturbance, such as ionospheric storm. So it is better to give some analysis about the availability of such TEC data for deriving the tidal components in the study.**

*We agree with the referee's concern regarding the uneven distribution of GNSS stations and the potential limitations of the TEC data over oceanic regions. The other referee has similar comments. To address this, we have added the following description to the Data and Methodology section:*

It should be pointed out that the GNSS stations are unevenly allocated, especially in earlier periods. Over vast oceanic regions near the equator GNSS receivers were sparsely set up on islands where adjacent receivers separated by a longitude difference up to 20 degrees. There were no receivers in the

Southern hemisphere high latitudes around 120°W over the Western Pacific Ocean and 15°W over the Atlantic Ocean (Schaer, 1999). Additionally the inclination of GNSS satellites inherently limits the satellite visibility at high latitudes near the polar region. In areas lacking observation the TEC retrieval inevitably involves interpolation, which can affect the accuracy. Therefore, our analysis focuses on low and mid-latitudes, where GNSS data is more reliable.

*We have added the reference,*

Schaer, S. Mapping and Predicting the Earth's Ionosphere Using the Global Positioning System. Ph.D. Thesis, Bern University, Bern, Switzerland, 1999.

**2,What is the criteria for the SSW events in Table 1 as MAJOR?**

**In general, the major SSW event mainly occurs in the winter period of the Northern Hemisphere, which is manifested by the reversal from eastwards to westwards of the zonal wind zonal mean and the increase of the stratospheric temperature in the polar region.**

**How is the Central date determined, and is it the same time as the SSW onset in Figure 4? The date of the SSW event in 2010 and 2020 is 20100323 and 20200322. Strictly speaking, these two events should be classified as Final Warming, and the background condition of the zonal wind zonal mean during this kind of warming is different with the normal SSW event occurred in winter period.**

*The major SSW events in our study were selected based on the criteria defined by Goncharenko et al. (2021) and Hocke et al. (2024), which include the reversal of the stratospheric zonal wind at 10 hPa (around 30 km altitude) and at 60°N latitude. The central date of each SSW event (SSW onset) is determined by the time of this wind reversal, as provided by Palmeiro et al. (2023) using ECMWF reanalysis data. We have added the following description for Table 1 in the Data and methodology section:*

The central date of each SSW event is determined by the time when the zonal mean wind changes from eastward to westward at 10 hPa, northward of 60°N (Palmeiro et al., 2023; Vargin et al., 2022). The events dated 20100323 and 20220322 occurred later in the season. They could be classified as Final Warmings. However, they were included in our analysis because they met the criteria for major SSWs as defined by Goncharenko et al. (2021).

*In the review paper by Goncharenko (2021), the SSW definition and classification have been discussed in detail. Regarding major SSWs, they occur most often in the months of January and February; displacement events*

*occur throughout the period from November to March. So regarding the events in 2010 and 2020, we acknowledge that these events occurred later in the season and could be classified as Final Warmings. However, they were included in our analysis because they met the criteria for major SSWs as defined by Goncharenko et al. (2021). To clarify this, we have added the following text to the Introduction section:*

Although the specific definition of SSW has varied over years, it is now widely accepted that a major SSW event mainly occurs in the winter period of the Northern Hemisphere. It is manifested by the reversal of the stratospheric polar vortex from eastward to westward and an increase in the stratospheric temperature in the polar region (Goncharenko et al., 2021).

*To be clear and straightforward, we have added "major" with SSW in the abstract.*

*We have also added the following references in the revised manuscript:*
Palmeiro, F. M., García-Serrano, J., Ruggieri, P., Batté, L., and Gualdi, S.: On the Influence of ENSO on Sudden Stratospheric Warmings, J. Geophys. Res.-Atmos., 128, e2022JD037607, https://doi.org/10.1029/2022JD037607, 2023.
Vargin, P.N.; Koval, A.V.;Guryanov, V.V. Arctic Stratosphere Dynamical Processes in the Winter 2021–2022. Atmosphere 2022, 13, 1550. https://doi.org/10.3390/atmos13101550.

**3,The input layer of the neural network algorithm shown in Figure 1 only takes into account the annual and diurnal variations of the ionosphere. Why doesn't consider the seasonal variation, i.e., the 180-day period variation? The selected SSW events happened in the northern winter period, whether this collection of input layer without seasonal variation component affect the final results?**

*We thought on the direct effects from the sun and inevitable influences due to the earth rotation and revolution. The model fits to the global TEC observation with a zero systematic error and an root mean square error (RMSE) of 3.221 TECU, which is similar to the zero systematic error and the RMSE of 3.387 TECU for the modeling with global TEC from 1999 to 2011 in Mukhtarov et al. (2013a).*

*Accepting your suggestion we have re-run the neural network algorithm by adding the 180-day period variation to the input layer. The RMSE was improved from 3.221 TECU to 2.780 TECU, indicating a better fit to the observed TEC data. The main results of our composite analysis remained consistent. The enhancement after SSW onset keeps similar. We have*

*updated all figures in the revised manuscript.*

**4,Why the sites of diurnal tidal and semidiurnal tidal component given in figure 3 are different? It may be better to give the diurnal and semidiurnal tidal components during the same event at same site.**

*We thank the referee for this suggestion. We have revised Figure 3 (Figure 4 in the revised manuscript) to show both the diurnal and semidiurnal tidal components during the February 1999 SSW event at the same site (30°N, 105°E). This change provides a clearer comparison of the tidal variations at a single location.*

**5, Figure 6 and Figure 7 give the latitudinal distribution of the diurnal and semidiurnal components at certain meridian line during SSW period, respectively. Why are there no results for the same longitude? It is suggested to provide the latitudinal distribution of the diurnal and semidiurnal component at each meridian lines. In addition, little observational data in the southern hemisphere in the sector of 80 E meridian line, and the map data in this region is basically interpolated, the result in this meridian line is it reliable?**

**By the way, the vertical line in each figure is not clear.**

*We agree that the reliability of the TEC data over the -80°E meridian is not so good due to less ionospheric piercing points compared with -75°E. Though the results can be found similar from the following plots, we substitute -80°E with -75°E where the satellite visibility is better.*

*We have added the latitudinal distribution of the semidiurnal components at 90°E and the diurnal ones at -75°E to provide a more comprehensive analysis. The vertical lines in the figures have been made clearer.*

[Figure]

[Figure]

6, In the discussion section, it is necessary to analyze why the semi-diurnal tides in the mid-latitudes of the Southern Hemisphere are stronger than those in the mid-latitudes of the Northern Hemisphere, and what are the possible mechanisms. In addition, it has been suggested that the semi-diurnal tidal component is probably related to the enhanced semi-lunar tidal (M2) during the SSW, and the discussion about the M2 component enhancement in the Northern and Southern Hemisphere during SSW should be added in the discussion section. The following related papers can be referred.

Goncharenko, L. P., Harvey, V. L., Randall, C. E., Coster, A. J., Zhang, S.-R., Zalizovski, A., et al. (2022). Observations of Pole-to-Pole, Stratosphere-to-Ionosphere Connection. Frontiers in Astronomy and Space Sciences, 8, 768629. https://doi.org/10.3389/fspas.2021.768629

Liu, J., Zhang, D., Goncharenko, L. P., Zhang, S., He, M., Hao, Y., & Xiao, Z. (2021). The latitudinal variation and hemispheric asymmetry of the ionospheric lunitidal signatures in the American sector during major Sudden Stratospheric Warming events. Journal of Geophysical Research: Space Physics. https://doi.org/10.1029/2020ja028859

Jing Liu, Donghe Zhang, Shuji Sun, Yongqiang Hao, Zuo Xiao, Ionospheric Semidiurnal Lunitidal Perturbations During the 2021 Sudden Stratospheric Warming Event: Latitudinal and Inter‐Hemispheric Variations in the American, Asian‐Australian, and African‐European Sectors, Journal of Geophysical Research: Space Physics, 10.1029/2022JA030313, 127, 9, (2022).

*We have added the following text to the Discussion section:*
Several SSW event studies have highlighted that semidiurnal tides in the Southern hemisphere mid-latitudes, particularly around -75°E in the American sector, are stronger than those in the Northern hemisphere. This hemispheric asymmetry may arise from the amplification of lunar semidiurnal (M2) tides

during SSWs, which is the most pronounced in the American sector (Goncharenko et al., 2021; Liu et al., 2021; 2022). Additionally, the inclination angle of Earth's magnetic field lines in the Southern hemisphere mid-latitudes is smaller than in the Northern hemisphere, leading to more ionospheric TEC variations in the F-region due to electric field effects (Goncharenko et al., 2022).

*Goncharenko er al., 2022 shows observational evidence that SSW events generate truly global disturbances that reach the high latitudes of the opposite hemisphere. We also cite it in the Introduction to highlight the global effects of SSWs.*

*We also cited Liu et al. (2021, 2022) in the Introduction section.*

To referee2

We sincerely thank the referee for their insightful and constructive comments, which have been invaluable in enhancing the quality and clarity of our manuscript. We deeply appreciate the time and effort they have dedicated to reviewing our work and providing thoughtful suggestions.

Below, we provide detailed responses (in italic font style) to each of the referee's comments (in bold face). The changed text is in a normal font style.

**The paper presents a composite analysis of ionospheric response to multiple sudden stratospheric warmings. To isolate SSW response, the authors first develop empirical model of total electron content, and use data-model differences to see SSW effects. Composite analysis of 18 SSW events is the novel aspect of the paper. Global ionospheric variations and complex latitudinal and longitudinal patterns are also new and interesting features.**

**Overall, the Introduction is pretty weak and does not mention several important studies that describe the state of knowledge on the topic. The new empirical model that uses machine learning approach is an interesting development. However, it would be important to demonstrate the performance of the model and discuss several performance metrics, so that the reader can be more comfortable about the attribution of the observed effects to SSW and not to the model itself. The paper would also benefit from a more extended discussion of potential mechanisms responsible for the observed features. Overall, the paper is an interesting development and will be stronger after addressing several comments. Most of them are clarifications and should not be hard to address. I recommend a minor revision.**

**Major comments**

**L. 26-28 - As there is a lot of literature on SSWs, a better reference is needed here. For example, recent review of Baldwin et al., 2021 (see suggested references below).**
*We have revised the Introduction section to include a broader context of SSW-ionosphere coupling. The review by Baldwin et al. (2021) is now cited to frame the state of knowledge (Lines 35–38). Chau et al. (2009) and Goncharenko and Zhang (2008) are cited to point out that the related topic is comparatively a new field.*

**L. 32-33 - it has been established through multiple simulations that wind and temperature changes in the middle atmosphere are the primary**

**reasons for the amplification of tidal modes, not mesospheric polar vortex. Please revise the Introduction.**

*We have revised it by removing the "a change of the mesospheric polar vortex" and adding* "wind and temperature changes in the middle atmosphere" *in the Introduction section.*

**L. 55+ - there were several other studies that investigated response to SSW at middle to high latitudes, including for multiple SSW events - for example, Liu et al., 2021. The paper would benefit from a more comprehensive description of what is known.**

*We thank the referee for this important point. We have added the following text in the Introduction section:*

There were several other studies that investigated response to SSW at middle to high latitudes, including for multiple events. It has been shown that enhanced semidiurnal lunitidal (M2) perturbations extended to middle latitude in the Southern hemisphere. In the American sector around -75°E, semidiurnal tides in the mid-latitudes of the Southern hemisphere are stronger than those in the Northern (Liu et al., 2021; 2022).

**The GNSS receiver coverage substantially varies with latitude and longitude, and also varies in time, with earlier data containing fewer stations and hence using more interpolations. The study needs to reflect that and discuss potential implications on the results.**

*We agree with the referee's concern regarding the GNSS receiver coverage and their impacts on the TEC accuracy. The other referee has similar comments. To address this, we have added the following description to the Data and Methodology section:*

It should be pointed out that the GNSS stations are unevenly allocated, especially in earlier periods. Over vast oceanic regions near the equator GNSS receivers were sparsely set up on islands where adjacent receivers separated by a longitude difference up to 20 degrees. There were no receivers in the Southern hemisphere high latitudes around 120°W over the Western Pacific Ocean and 15°W over the Atlantic Ocean (Schaer, 1999). Additionally the inclination of GNSS satellites inherently limits the satellite visibility at high latitudes near the polar region. In areas lacking observation the TEC retrieval inevitably involves interpolation, which can affect the accuracy. Therefore, our analysis focuses on low and mid-latitudes, where GNSS data is more reliable.

*We have added the reference,*

Schaer, S. Mapping and Predicting the Earth's Ionosphere Using the Global Positioning System. Ph.D. Thesis, Bern University, Bern, Switzerland, 1999.

**Table 1 presents central dates of SSWs. As there are multiple ways of**

**defining a central day of SSW, exact dates (and hence the results of the study) can depend on the definition of central date. Please provide more details how central date was defined for this study.**

*We have added the following text in the Data and Methodology:*

The central date of each SSW event is determined by the time when the zonal mean wind changes from eastward to westward at 10 hPa, northward of 60°N (Palmeiro et al., 2023; Vargin et al., 2022). The events dated 20100323 and 20220322 occurred later in the season. They could be classified as Final Warmings. However, they were included in our analysis because they met the criteria for major SSWs as defined by Goncharenko et al. (2021).

**Development of empirical model of TEC is an important effort that can provide background TEC for a variety of other studies. It is important to understand how good is the model and how well it describes seasonal and solar cycle variations. The paper needs to include at least some examples of this, and to include several metrics evaluating the performance of the model. If the authors are reluctant to include them in the body of the paper, they can be included as Attachment.**

*Thank you for pointing out this important point. We have provided the systematic error and root mean square error (RMSE), and added a plot of maps for observed and modelled TEC (Figure 2 in the revised manuscript) in the Data and Methodology section.*

**There were several earlier efforts to develop empirical TEC models using the same (although shorter) TEC dataset. For example, Mukhtarov et al., 2013a, b; Lean et al., 2016. They need to be mentioned for the sake of scientific objectivity. How does the model developed in this study perform compared to the earlier models?**

*Following referee's suggestion, we have made comparison and added the following text to the Data and Methodology section:*

The ML-TEC model fits to the global TEC observation with a zero systematic error and a root mean squares error (RMSE) of 2.8 TECU. This is comparable to the zero systematic error and the RMSE of 3.4 TECU for the empirical funntion modeling with the global TEC from 1999 to 2011 in Mukhtarov et al. (2013), and the RMSE of 3.5 TECU for a statistical model established by Lean et al. (2016) with the global TEC from 1998 to 2015. Figure 2 presents the global maps of the modeled and observed TEC in geographical coordinate. The equatorial ionospheric anomaly (EIA) locates between 22.5°S and 25°N around 105°E, with the summer crest being stronger than the winter one. The Weddell Sea Anomaly is apparent with the stripe amplification between 80°S to 50° S and -120°E to 0° E (Mukhtarov et al., 2013). The coincidence of these anomalies indicates the ML-TEC model is also able to reproduce the spatial structure of the ionosphere.

[Figure]

Figure 2. Global maps of the modeled and observed TEC at 0800 UT on 12 December 2012. The lines in magenta represents the magnetic equator.

**In Figure 3, diurnal and semidiurnal components are given for the same latitude but different longitudes. What is the justification for this?**
*We just wanted to show results at more places. The other referee also has the same point. We have revised Figure 3 (Figure 4 in the revised manuscript) to show both the diurnal and semidiurnal tidal components during the February 1999 SSW event at the same site (30°N, 105°E). This change provides a clearer comparison of the tidal variations at a single location.*

**In addition, the authors attribute all the data/model differences to SSW. However, largest post-SSW difference of ~5TECu coincides with increase in solar flux due to the 27-day solar rotation, and some of the differences could be potentially attributed to the model performance for different seasons and solar flux levels. This is why it is important to present some evidence of model performance, per my earlier comment.**
*We realize we should be careful with the description in English, and don't rush to the attribution to SSW at this stage.*

**In Figure 4, what is the justification for showing delta S1 at 13 days before SSW? Are you implying that SSW effects start 13 days before the central date? Are these patterns statistically significant?**
*Since both the diurnal and semidiurnal components vary with latitude, longitude and time, we select world maps of those with overall smallest values before SSW onset and largest values after SSW onset. We have added the following text before presenting Figure 4 (Figure 5 in the revised manuscript):*

Results of composite analysis are shown by world maps, latitude-time and longitude-time plots. Since both the diurnal and semidiurnal components vary with latitude, longitude and time, we select world maps of those with overall smallest values before SSW onset and largest values after SSW onset. For clarity in our description and discussion, we use hereafter the subscript "b" to denote the period before the SSW onset, and the subscript "a" to denote the period after the SSW onset.

**Describing Figure 4, the authors write 'The largest deltaS1 enhancement is 2.25 TECU and locates at (2.5°S, 90°W)', and several lines later they write 'Largest deltaS1 is ~1.95 TECU and locates at 2.5N and [45°W, 50°W]'. Please clarify the meaning of this - it is not clear what the authors are trying to emphasize.**
*In the revised manuscript, we define the enhancement to be the difference between those after SSW onset and before SSW onset. We use subscript "E" to denote the enhancement.*

$$\Delta S_{1E} = \Delta S_{1a} - \Delta S_{1b}$$
$$\Delta S_{2E} = \Delta S_{2a} - \Delta S_{2b}$$

*We modified the descriptions correspondingly.*

**Similar comments about Figure 5 - figure 5a shows distribution of deltaS2 for 12 days before the SSW onset. Why 12 days? Why not 10 days or 15 days, and why this is different from 13 days before SSW onset for Figure 4? Are these variations statistically significant? How do they compare with, for example, 1 sigma or RMSE for the model?**

*The temporal variation of semidiurnal is from diurnal. Its world map shows overall smallest values at 12 days before SSW onset. Because it varies with position and time, we inspected world maps from -30 to 0 days and selected the overall smallest one.*

**Figure 6 shows deltaS1 at one longitude. Please say few words whether patterns are similar or substantially different at other longitudes.**
*For this figure 6 (Figure 7 in the revised manuscript), we have added plot of deltaS2. We added another longitude of -75E to the next figure (Figure 8 in the revised manuscript). So the patterns can be found different at 90E and -75E.*

**Figure 7 shows deltaS2 at a different longitude, 80W. Why is it different from the longitude in Figure 5? Why are these specific longitudes selected?**
*We just wanted to show results at more places. As the other referee points out less reliable TEC at 80W, we substitute 80W with 75W and added deltaS1 at 75W in the figure (Figure 8 in the revised manuscript).*

*We have added the following text before presenting Figures 6 and 7 (Figures 7 and 8 in the revised manuscript):*

It is important to examine the ionospheric tidal variabilities at different longitudes over time. We select two longitudes of 90°E and 75°W. As shown above by Figures 5 and 6, at ~90°E there is obvious enhancement of $\Delta S_1$ in northern mid-latitudes; at 75°W prominent SSW effects can be seen in both hemispheres.

**For all figures 4-9 (or at least for some of them), it might be worthwhile to add another panel that shows variations not in absolute units of TEC, but as percentage compared to the background (model). This might help to illuminate the relative strength of SSW-related disturbances at different latitudes and longitudes.**

*We thank the referee for this suggestion. We use rS to represent the ratio of the S from observed to the S from modeled TEC. We have added rS1 and rS2 in the last block of the flowchart in Figure 2 (Figure 3 in the revised manuscript), and the following text to the flowchart description.*

As shown by $rS_1$ and $rS_2$, the ratios of those observed to the modeled ones are also calculated to show the relative strength of SSW-related disturbances.

*We have plotted world maps of rS1 and rS2 and added them in another panel beside those of DeltaS1 (Figure 5 in the revised manuscript) and Delta S2 (Figure 6 in the revised manuscript). The spatial patterns of relative strengths of SSW-related disturbances is similar to those of DeltaS1 and DeltaS2. We have added the following text in the Results section:*

The right panel of Figure 5 is for $rS_1$ with (c) at 13 days before SSW onset and (d) at 25 days after SSW onset. The $rS_1$ larger than 1 matches positive $\Delta S_1$, and the $rS_1$ smaller than 1 to negative $\Delta S_1$. Similar spatial distributions can be noticed to those of $\Delta S_1$ by comparing the corresponding maps in the left panel. At 25 days after SSW onset $rS_{1a}$ is also stronger in the Northern hemisphere than the Southern hemisphere. However, it has a similar level at the Northern low and mid-latitudes. Note that largest $rS_1$ locates at high latitudes near polar regions, which is different from that of $\Delta S_1$. This can be attributed to the small values of diurnal variation due to the smaller TEC there than low to mid-latitudes.

The right panel of Figure 6 shows $rS_2$ with (c) at 12 days before SSW onset and (d) at 8 days after SSW onset. There are also similar relationships between $rS_2$ and $\Delta S_2$ by comparing the corresponding maps in the two panels. At 8 days after SSW onset, $rS_{2a}$ is obviously stronger in the Southern hemisphere than the Northern hemisphere.

**Figure 8 shows interesting longitudinal features. The study needs to include a discussion of potential reasons for these variations.**
*We have added discussion on longitudinal feature and potential reasons in the Discussion section.*

**Figure 9 - same comment as earlier; 20N is selected for figure 8, but 22.5N for figure 9. Why?   As the study uses TEC maps with latitude grid of 2.5 degrees, differences at 20N and 22.5N should not be large.**
*Yes, differences are not large. We have revised Figures 8 and 9 into one figure (Figure 9 in the revised manuscript) to show the diurnal and semidiurnal tidal components at 22.5N.*

**In figure 9, the authors emphasize enhancement at 45-135E (note also, there is a typo there, should be 45-135E, not 45-135N). But enhancement is also seen around day -50 to -40. How confident are you that enhancements after the SSW onset in that longitude range can be truly attributed to SSW, and not to, for example, insufficient data coverage at these longitudes?**
*We realize we should be careful with the description in English. Actually longitudinal variability is more or less different at different latitudes concerning the value, duration and changing rate of the $\Delta S_1$ and $\Delta S_2$. Regarding the enhancement seen around -50 or -40 days before SSW onset, the value, duration and changing rate of $\Delta S_1$ are all smaller than those after SSW onset in the Northern hemisphere; they can be smaller and with less organized pattern after SSW onset, particularly in low latitudes in the Southern hemisphere. For those of $\Delta S_2$, generally they are all smaller than those after SSW onset. We have added the following text to the Results section:*

We notice that positive $\Delta S_1$ and $\Delta S_2$ also occur and last for days before SSW onset. However, they are generally smaller, shorter-lived and

slower-varying than those after SSW onset.

**Overall, the Discussion section is pretty weak and could benefit from more extended discussion about the potential mechanisms for the observed features and comparison with available studies.**

*We thank referee for the suggestion. We have extended the discussion and added the following text to the Discussion section:*

Several SSW event studies have highlighted that semidiurnal tides in the Southern hemisphere mid-latitudes, particularly around -75°E in the American sector, are stronger than those in the Northern hemisphere. This hemispheric asymmetry may arise from the amplification of lunar semidiurnal (M2) tides during SSWs, which is the most pronounced in the American sector (Goncharenko et al., 2021; Liu et al., 2021; 2022). Additionally, the inclination angle of Earth's magnetic field lines in the Southern hemisphere mid-latitudes is smaller than in the Northern hemisphere, leading to more ionospheric TEC variations in the F-region due to electric field effects (Goncharenko et al., 2022).

Recently, Harvey et al. (2022) emphasized the influence of the mesospheric polar vortex on atmospheric tides, which helps explain this hemispheric asymmetry. Since major SSWs occur predominantly in the Northern Hemisphere (NH) during winter, the mesospheric polar vortex in the NH significantly modulates the upward propagation of atmospheric tides to the ionosphere. This process enhances the diurnal variation of TEC, making it more pronounced in the Northern Hemisphere.

Harvey V. L., Randall C. E., Bailey S. M., Becker E., Chau J. L., Cullens C. Y., Goncharenko L. P., Gordley L. L., Hindley N. P., Lieberman R. S., Liu H-L, Megner L., Palo S. E., Pedatella N. M., Siskind D. E., Sassi F., Smith A. K., Stober G., Stolle C. and Yue J.: Improving ionospheric predictability requires accurate simulation of the mesospheric polar vortex. Front. Astron. Space Sci. 9:1041426. doi: 10.3389/fspas.2022.1041426, 2022.

**As empirical model takes substantial time to develop and can be used for other studies as a background, it would be important to provide access to model output code to the reader, as currently expected in different journals.**

*We would like to share the code and the modeled TEC data, which are mentioned in the code availability and data availability in the revised manuscript.*

**Acknowledgment mentions foF2 data for Okinawa and Wuhan, which is not relevant to this study.**

*Many thanks! We have deleted this sentence.*

**Minor comments & language**

**L. 10 - 'SSW effect is mainly in low-latitude ionosphere' - SSW effects are observed mainly in the low-latitude ionosphere**
*We have changed it.*

**L. 21 - 'lasts to about 50 days after SSW onset' - lasts for about 50 days after SSW onset?**
*Yes, we have changed it to* "lasts for", and "about 20~50 days after SSW onset" *according to Figures 7-9.*

**It is better to avoid using abbreviations in the abstract, and introduce abbreviations the first time they are used. For example, 'SSW' is used in the abstract, but not defined.**
*We have added (SSW) after the sudden stratospheric warming mentioned the first time.*

**Some references are missing in the reference list - for example, Chau et al., 2009, Goncharenko et al., 2018; Yamazaki et al., 2012. Please check the references list carefully.**
*We have added the references in the reference list.*

**L. 120 - 'only those driven by the atmosphere below are remained' —> 'only those driven by the atmosphere below are retained' or 'only those driven by the atmosphere remain'**
*We have change the phrase to* "only those driven by the atmosphere below are retained".

**L. 151 - 'In southern atmosphere' —> In southern hemisphere?**
*We have revised it.*

**Vertical line that marks SSW onset in figures 6-9 could be made thicker, it is barely seen now.**
*We have made the vertical lines thicker in the revised manuscript.*

**L. 235 - 'is larger Northern hemisphere' —> 'is larger in the Northern hemisphere'**
*We have revised it accordingly.*
* * *

[Figure]

Figure 2. Global maps of the modeled and observed TEC at 0800 UT on 12 December 2012. The lines in magenta represents the magnetic equator".

9. Replaced "remained" in the last paragraph with "retained".

10. Added in the end of last paragraph "As shown by $rS_1$ and $rS_2$, the ratios of those observed to the modeled ones are also calculated to show the relative strength of SSW-related disturbances".

11. Figure 2 was renameed as Figure 3. Added "Ratio of composites (Observation/Model) rS1, rS2" in the bottom block of Figure 3.

Changes in Results:
1. Figure 3 was renamed as Figure 4. The relevant change in Figure 4 and hence the description are the following:
   (1) the maximum difference between observed and modeled s1 is 2.5 TECU (previously 5 TECU); for s2 it's 1.6 TECU (previously 3.6 TECU).
   (2) the period (observed s1 larger than the modeled one) lasts for ~30 days

(previously 50 days).

2. Added "Results of composite analysis are shown by world maps, latitude-time and longitude-time plots. Since both the diurnal and semidiurnal components vary with latitude, longitude and time, we select world maps of those with overall smallest values before SSW onset and largest values after SSW onset. For clarity in our description and discussion, we use hereafter the subscript "b" to denote the period before the SSW onset, and the subscript "a" to denote the period after the SSW onset" at the beginning of the second paragraph.

3. Figures 4 and 5 were renamed as Figures 5 and 6, respectively.

4. Defined the "enhancement" to be the difference between those after SSW onset and before SSW onset. We use subscript "E" to denote the enhancement.

$$\Delta S_{1E} = \Delta S_{1a} - \Delta S_{1b}$$
$$\Delta S_{2E} = \Delta S_{2a} - \Delta S_{2b}$$

5. For figure 5 in the revised manucript,
   (1) replace "The largest enhancement is 2.25 TECU and locates at (2.5°S, 90°W)" with "The largest $\Delta S_{1E}$ is 1.5 TECU and locates at (5°N, 85°W)".
   (2) replace the description for Figure 5b with "Shown in Figure 5b, at 25 days after SSW onset, conspicuous positive $\Delta S_{1a}$ prevails almost globally. Negative $\Delta S_{1a}$ occurs at low latitudes from the Atlantic Ocean to the Indian Ocean in the Southern hemisphere and mid-latitudes in Pacific Ocean. While positive $\Delta S_{1a}$ shows the largest at the EIA, and moderate strength around 90°E and 75°W in mid to high latitudes in the Northern hemisphere, it occupies mid to high latitude for all longitudes in the Southern hemisphere with positive patches as well in low latitudes in south America and its west".
   (3) Added at the middle of the paragraph describing Figure 5 "Contrasting $\Delta S_{1a}$ and $\Delta S_{1b}$, the amount of change, referred to as, $\Delta S_{1E}$ can be obtained by taking the difference, $\Delta S_{1a} - \Delta S_{1b}$. An enhancement (positive $\Delta S_{1E}$) can be discerned globally, which is generally stronger in

the Northern hemisphere than the Southern one. The largest enhancement mainly distributes in low northern latitudes in a longitude range of [155°W, 120°E]. The largest $\Delta S_{1E}$ is 1.5 TECU and locates at (5°N, 85°W). In the Southern hemisphere, $\Delta S_{1E}$ can be larger than 1.0 TECU from low to high latitudes around 75°W in American sector".

(4) Added another panel in figure 5, and added the following text at the end of the paragraph describing results shown in the right panel of figure 5 "The right panel of Figure 5 is for $rS_1$ with (c) at 13 days before SSW onset and (d) at 25 days after SSW onset. The $rS_1$ larger than 1 matches positive $\Delta S_1$, and the $rS_1$ smaller than 1 to negative $\Delta S_1$. Similar spatial distributions can be noticed to those of $\Delta S_1$ by comparing the corresponding maps in the left panel. At 25 days after SSW onset $rS_{1a}$ is also stronger in the Northern hemisphere than the Southern hemisphere. However, it has a similar level at the Northern low and mid-latitudes. Note that largest $rS_1$ locates at high latitudes near polar regions, which is different from that of $\Delta S_1$. This can be attributed to the small values of diurnal variation due to the smaller TEC there than low to mid-latitudes."

6. For figure 6 in the revised manuscript,
   (1) Replaced "2.2 TECU" with "2.0 TECU around (30°S, 85°W)" for the largest $\Delta S_{2a}$ .

   (2) Replaced "The largest enhancement is ~1.85 TECU at ~20°S in Pacific Ocean" with "Near south America the largest $\Delta S_{2E}$ is 1.5 TECU at (32.5°S, 80°W) though it can reach 1.8 TECU around $\pm$~20°N over Pacific Ocean".

   (3) Added a panel in figure 6, and describe the results as "The right panel of Figure 6 shows $rS_2$ with (c) at 12 days before SSW onset and (d) at 8 days after SSW onset. There is also similar relationships between $rS_2$ and $\Delta S_2$ by comparing the corresponding maps in two panel. At 12 days

after SSW onset, $rS_{2a}$ is obviously stronger in the Southern hemisphere than the Northern hemisphere".

7. Figure 7 in the revised manuscript,
   (1) Added "We select two longitudes of 90°E and 75°W to examine the temporal variations of $\Delta S_1$ and $\Delta S_2$. As shown above by Figures 5 and 6, at ~90°E there is obvious enhancement of diurnal component in northern mid-latitudes; at 75°W prominent enhancement of both diurnal and semidiurnal can be seen in both hemispheres."
   (2) Replaced "From ~5°N to ~60°N" with "At ~25°N".
   (3) Replaced "-5 to 35" with " ~15 to ~35°N".
   (4) Replaced "At ~30°N, DeltaS1 shows a peak level of ~1.4 TECU around 25 days. At ~15°N, $\Delta S_1$ maintains …" with "$\Delta S_1$ shows a peak level of ~0.6 TECU around 10 days, and maintains the highest level from 0 to ~35 days after the SSW onset".
   (5) Added latitude and temporal variation of semidiurnal variation. Added "Concerning the semidiurnal component, clear enhancement of $\Delta S_2$ starts just from the SSW onset in the Northen hemisphere. The enhancement ends at ~35 days in mid-latitudes and ~50 days at low to mid-latitudes. The largest is 1.3 TECU centering ~20°N and ~8 days after SSW onset".

8. Figure 8 in the revised manuscript,
   (1) Replaced $\Delta S_2$ of 80°W with that of "75°W".
   (2) Added $\Delta S_1$ of 75°W and the description "It is obvious that from ~0°N to ~70°N an enhancement of $\Delta S_1$ starts from the SSW onset, it ends at ~50 days in low latitudes and at~30 days in mid to high latitudes. At ~2.5°N $\Delta S_1$ increases very fast and maintains a high level from ~8 to 22 days with a largest value of 0.8 TECU. In the southern hemisphere the enhancement of $\Delta S_1$ delays with the latitude to ~30°S".
   (3) Replaced "At ~5°N and ~30°S" with "Between ~30°S and ~5°N".
   (4) Replaced "first peak of ~1.9 TECU" with "first peak of ~1.5 TECU".

9. Figure 9 in the revised manuscript,
   (1) Replaced "$\Delta S_1$ at 20°N" with "$\Delta S_1$ at 22.5°N", added in figure 9.

(2) Replaced "The $\Delta S_1$ is generally enhanced from 0 to 50 days after the SSW onset for all longitudes except ~25 W where $\Delta S_1$ starts to decrease from ~30 days after the SSW onset" with "The $\Delta S_1$ generally starts to increase at the SSW onset and keeps positive for ~30 days".

(3) Replaced "20-30 days" with "20-26 days".

(4) Replaced "a maximum of ~1.5 TECU" with "a maximum of ~1.4 TECU at ~120°E".

(5) Replaced "45-135°N" with "60-135°E".

(6) Replaced "$\Delta S_2$ at the other area decreases to" with "$\Delta S_2$ between -120-60°E decreases to".

(7) Replaced "At areas of -150 to -40°E and -10 to 45°E, $\Delta S_2$ is at a level of ~0, and starts to increase at the SSW onset, reaches a maximum at ~10 days and decrease to a low level at ~30 days after the SSW onset" with "Over the Western Pacific Ocean it returns to the onset level at ~20 days".

(8) Added "We notice that positive $\Delta S_1$ and $\Delta S_2$ also occur and last for days before SSW onset. However, they are generally smaller, shorter-lived and slower-varying than those after SSW onset".

Changes in Discussion:
1. Added "and temporal variations over latitude/longitude" after "The world maps" in the beginning of the third paragraph.
2. Added "Several SSW event studies have highlighted that semidiurnal tides in the Southern hemisphere mid-latitudes, particularly around -75°E in the American sector, are stronger than those in the Northern hemisphere. This hemispheric asymmetry may arise from the amplification of lunar semidiurnal (M2) tides during SSWs, which is the most pronounced in the American sector (Goncharenko et al., 2021; Liu et al., 2021; 2022). Additionally, the inclination angle of Earth's magnetic field lines in the Southern hemisphere mid-latitudes is smaller than in the Northern hemisphere, leading to more ionospheric TEC variations in the F-region due to electric field effects (Goncharenko et al., 2022)".
3. Replaced "Longitudinal differences exist in both diurnal and semidiurnal tides. Weaker amplification is obvious in Atlantic, African and Indian Sectors. This is especially true for the diurnal tide in Southern hemisphere" with "This contrasts with the semidiurnal enhancement, which is stronger in the Southern hemisphere. This contrasts with the semidiurnal enhancement, which is stronger in the Southern hemisphere. Longitudinal differences are

also evident, with weaker amplification in the Atlantic, African, and Indian sectors, particularly for the diurnal tide in the Southern hemisphere" in the fourth paragraph.

4. Added "Recently, Harvey et al. (2022) emphasized the influence of the mesospheric polar vortex on atmospheric tides, which helps explain this hemispheric asymmetry. Since major SSWs occur predominantly in the Northern Hemisphere (NH) during winter, the mesospheric polar vortex in the NH significantly modulates the upward propagation of atmospheric tides to the ionosphere. This process enhances the diurnal variation of TEC, making it more pronounced in the Northern Hemisphere".

5. Replaced "As for the temporal variation due to atmospheric force below, DeltaS1 at 22.5°N starts to increase simultaneously on the day of SSW onset in figures 6 and 8. It reaches to the peak around 20-30 days after SSW onset"

with "Temporally the global diurnal component ($\Delta S_1$) at 22.5°N starts to increase simultaneously on the day of SSW onset, peaking around 20–26 days later and lasting for approximately 30 days (Figures 9)".

6. Replaced "both diurnal and semidiurnal components last for ~50 days" with "the diurnal and semidiurnal components last for ~30 and ~50 days, respectively".

7. Replaced "not strong enough to produce vertical plasma drifts in high-latitude" with "larger in the low-latitude ionosphere to produce more significant vertical plasma drifts than mid-latitudes".

Changes in Conclusion:
1. Added "Key findings include:" and categorize "The semidiurnal TEC variation… after SSW onset" into four short paragraphs starting by the following, respectively.
2. Added "Semidiurnal TEC Variations".
3. Added "Diurnal TEC Variations".
4. Added "Temporal Evolution".
5. Added "Longitudinal Dependence".

Changes in data availability:
Added "The modelled TEC data can be provided upon request".

Changes in References:
Added the following references,

[revised manuscript text omitted]